# High-Intensity Pulse Magnetic Fields Affect Redox Homeostasis and Survival Rate of *Escherichia coli* According to Initial Level of Intracellular Glucose

**DOI:** 10.3390/biom15111550

**Published:** 2025-11-05

**Authors:** Pengbo Wang, Limeng Du, Yunchong Li, Zitang Xu, Luona Ye, Shuhan Dai, Li Xu, Jinyong Yan, Xiaoman Xie, Quanliang Cao, Min Yang, Xiaotao Han, Yunjun Yan

**Affiliations:** 1Key Laboratory of Molecular Biophysics of the Ministry of Education, College of Life Science and Technology, Huazhong University of Science and Technology, Wuhan 430074, China; wangpb@hust.edu.cn (P.W.);; 2Wuhan National High Magnetic Field Center, Huazhong University of Science and Technology, Wuhan 430074, China

**Keywords:** magnetic field treatment, glucose metabolism, cell division, cell survival, NADH/NAD^+^ ratio, ROS

## Abstract

The biological effects of magnetic fields (MFs) have been studied and applied in medicine over the past four decades. However, the influence of high-intensity pulse magnetic fields (HI-PMFs), theorized to exert even stronger biological effects, is rarely reported. Herein, a study was conducted to investigate the biological effects of 2.5 T HI-PMF on the model organism *Escherichia coli* and its corresponding physiological alterations. After being treated by HI-PMF, a notable increase was observed in its intracellular NADH/NAD^+^ ratio, coupled with an improved cell survival rate. Transcriptome analysis revealed significant upregulation of genes related to glucose metabolism. Subsequent experiments confirmed that if the initial intracellular glucose level was relatively high and markedly decreased after being treated with HI-PMF, the cell density would significantly rise, owing to the alleviated inhibition of cell division. On the contrary, a lower initial intracellular glucose level led to cell death under HI-PMF. Furthermore, reactive oxygen species (ROS) production was proved to be the main cause attributed to the above phenomena. Therefore, our study suggests that HI-PMF treatment promotes ROS production, enhances cellular glucose metabolism, and consequently influences cell division and survival rate according to the initial level of intracellular glucose.

## 1. Introduction

Magnetic fields (MFs), such as the geomagnetic field (GMF) and electromagnetic field (EMF), are a persistent external stimulus for organisms, may have played a role in the emergence of life [1], and has been used for localization by some organisms [2,3]. At the present time, the emergence of various artificial MFs has led to interest in their safety [4,5], effects on cells [6,7], and application in medicine and biology [8,9,10]. Promising applications of MFs have been proposed in disease treatment [11,12], microbial fermentation [13,14], pollutant treatment [15,16], and so on. However, the influence of MFs on cells depends on their intensity and frequency [17,18,19]. The variability of these two factors makes it unclear how some types of MFs affect cells.

In addition, most of the existing studies are based on weak-intensity MFs (WI-MFs), whose magnetic flux densities are less than 1 Tesla (T). WI-MFs with different intensities and frequencies have been found to affect many phenotypes of cells, such as morphology [20,21], metabolism [22,23,24], signal transduction [25,26], and growth [19,27,28]. High-intensity MFs (HI-MFs), with a peak magnetic flux density greater than 1 T, are less researched. Related research mainly focuses on high-intensity static MFs (HI-SMFs), whose intensities are between 1T and 10T and do not change, due to their application in magnetic resonance imaging (MRI). These studies show that HI-SMFs affect *zebrafish larva* movements [29], improve mental health in mice [30,31], extend the lifespans of mice [32], and have anti-tumor potential [33,34]. At the cellular level, HI-SMFs affect cellular ATP levels [35], the orientation of mitotic spindles [36] and gene expression [37].

High-intensity pulse MFs (HI-PMFs) are another type of HI-MF that generate MFs intermittently. The rapid intensity changes in HI-PMFs may cause their effects on cells to be different than those of HI-SMFs. A study indicated that HI-PMF treatment at 8 T and 60 pulses destroyed cell membranes, caused cell content leakage, and induced cell death in *Escherichia coli* [38]. Similar cell death caused by HI-PMF treatment was reported in *Listeria monocytogenes* [39]. However, other studies show opposing results. An earlier study showed that HI-PMF treatment at 18 T and 50 pulses did not affect the survival of *E. coli*, even when combined with other stimuli [40]. Another study suggested that HI-PMF treatment at 6.7 T and 350 pulses caused no release of damage-associated molecular patterns in three animal cells [41]. The differences between different studies are obvious, and existing studies are too few to explain them. Furthermore, some studies only focused on cell survival rate and lack descriptions of cell physiology, making it difficult to summarize the mechanisms of the cellular effects of HI-PMFs. Therefore, it is imperative to conduct more systematic research of this aspect.

Some hypotheses have been proposed to explain the biological effects of MFs, such as different experiences of force of different cellular components in MFs due to their magnetic anisotropy [42], effects on cell membrane potential due to electromagnetic induction [43,44], and the radical pair mechanism (RPM) model [45]. Among them, RPM is the most widely accepted one, which indicates that MFs affect the stability of intracellular radical pairs [3,46,47]. In this way, MFs can alter the redox state of cells and lead to various effects [19,23,48,49]. However, it is unclear whether RPM contributes to the cellular effects of HI-PMFs.

To explore the cellular effects of HI-PMFs, this study aims to explore the redox changes in *E. coli* cells following treatment with a moderate intensity of 2.5 T HI-PMF. Unexpectedly, increases in the NADH/NAD^+^ ratio and cell count were observed, as well as a reduction in cell length after being treated by HI-PMF, which were not discovered by existing studies. Then the underlying mechanisms of these phenomena were explored. Transcriptome sequencing analysis was employed to identify the genes affected by HI-PMF, revealing the significant variations in many genes related to glucose metabolism. Because glucose metabolism influenced cell count by energy generation and division inhibition of *E. coli* cells [50], it was supposed that glucose storage played a key role in the physiological changes in *E. coli* cells after being treated by HI-PMF. This was demonstrated by subsequent experiments, which showed that different glucose supplies led to different experimental results. Finally, it was discovered that an increased metabolism was induced by the reactive oxygen species (ROS) generated by RPM, and appropriate ROS stimulation by H_2_O_2_ also could promote cell division in *E. coli*. Our study suggests that HI-PMF treatment enhances cellular glucose metabolism by promoting ROS production, and further, affects cell survival according to the initial level of intracellular glucose.

## 2. Materials and Methods

### 2.1. Strains and Sample Preparation

All strains used in this study were derived from *E. coli* BL21(DE3). Standard techniques were employed for cloning, electroporation, and other genetic manipulations. Crisp/Cas9 technology was used to insert the SoNar gene into the genome and knock out genes [51]. The Pet-28a plasmid was used to overexpress genes. The *E. coli* strains used are described in the Appendix A.

*E. coli* was cultured in Luria–Bertani (LB, 1% tryptone, 0.5% yeast extract, 1% NaCl) at 37 °C for 2.5 h and adjusted to 20 °C after adding IPTG (final concentration of 10 mmol/L) for 16 h. Then the culture was centrifuged, resuspended in an equal volume of sterile phosphate-buffer solution (PBS, pH 7.2), and placed on ice until HI-PMF treatment. The control group was kept in the same environment without HI-PMF treatment. Then the sample was placed on ice again until various tests were conducted. Three biological replicates were performed for each treatment condition.

### 2.2. HI-PMF Treatment

Pulsed magnetic field equipment was made by the school of electrical and electronic engineering, Huazhong University of Science and Technology. Fully charged capacitors discharged a current to the treatment chamber, which included a coil used to generate a MF. The maximum MF intensity generated by the coil was 3.5 T. The stimulus waveform was monophasic, with a pulse width of 6 ms and 30 s interval (Appendix A). It was paused for 5 min every 3 pulses to maintain the temperature at 20 degrees Celsius. The treatment chamber was cylindrical with 30 cm in height and 60 cm in diameter. Samples were placed inside the coil, which was 15 cm in height and 20 cm in diameter. The intensity of the PMF in the treatment chamber was spatially uniform and adjusted by the input voltage. First, 10 pulses of HI-PMF treatment with intensities of 0.5 T, 1.5 T, 2.5 T, and 3.5 T were processed. Then 2.5 T HI-PMF with pulse numbers from 5 to 30 were tested. Finally, 2.5 T and 15 pulses were chosen as the main HI-PMF parameters and used in subsequent experiments.

### 2.3. Starvation Stimulation and Glucose Stimulation

*E. coli* was cultured according to the above method. The starvation group (S group) and the glucose group (S + G group) were sampled two hours in advance and resuspended in PBS before HI-PMF treatment. The S group received no additional glucose in PBS, while the S + G group was supplied with 0.04% glucose. The residual culture continued to incubate in the LB broth until treatment. Then it was divided into two groups and resuspended in PBS. One of them was set as the control group (C group) and received no additional glucose, and the other (C + G group) received 0.04% glucose. The four groups were treated by HI-PMF as mentioned above. Three biological replicates were performed for each group.

### 2.4. H_2_O_2_ Stimulation

H_2_O_2_ with 30% mass concentration was sterilized through filtration and then diluted into 10 mol/L (M), 7.5 M, 5 M, 2.5 M, 2 M, 1 M as mother liquors. *E. coli* was cultured and resuspended in PBS according to the above methods. Then the *E. coli* in PBS was placed on ice for 30 min, similarly to processes in HI-PMF treatment. The resuspension was divided into 1 ml portions, and 10 μL of the H_2_O_2_ mother liquors with different concentrations were added. Then various tests were conducted. Three biological replicates were performed for each treatment condition.

### 2.5. Detection of Growth

The cells after being treated by HI-PMF were resuspended aseptically in an equal volume of LB. A total of 200 μL of the bacterial solution was added to a 96-well plate and cultivated in a 37 °C incubator. The OD_600_ was measured every hour by FlexStation3 (Molecular Devices, San Jose, CA, USA). Three biological replicates and three technical replicates were performed.

### 2.6. Cell Counting

Samples were serially diluted in PBS, and 100 μL of the dilution was planted on LB agar. The plates were incubated at 37 °C for 24 h. After incubation, the colonies were counted and the survival rate was expressed as N/N_0_, where N_0_ is the number of microorganisms in the control sample, and N is the number of microorganisms in PMF-treated sample. Three biological replicates were performed.

### 2.7. Image Collection and Cell Length Measurement

Fluorescent images were taken using the OLYMPUS FV3000 fluorescence stereomicroscope (Evident Scientific, Waltham, MA, USA). The wavelengths of excitation were 405 nm and 485 nm. Both emission lights were detected from 500 nm to 600 nm. The two excitation lights were quickly scanned in sequence to avoid mutual influence. The acquisition settings were the same for all images. Imagej (version 1.54d) was used to orient, scale, and merge images. The length of cells was calculated using the image software according to the fluorescent pictures. Data were neither added nor subtracted; original images are available upon request.

### 2.8. Fluorescence Intensity Detection

Some of the fluorescence intensities were measured according to the fluorescent images by Imagej, and the other fluorescence intensities were detected by FlexStation3 (Molecular Devices, USA). Which one was used for each of the different samples is described in the text. A total of 200 μL of the bacterial suspended in PBS was detected in a 96-well black plate. The wavelengths of excitation were 405 nm and 485 nm. Both emission lights were detected at 520 nm. During continuous testing, the 96-well plate was placed in FlexStation3 and tested every minute. Three biological replicates and three technical replicates were performed for each treatment condition.

### 2.9. RNA Extraction and Sequencing

*E. coli* with and without PMF treatment was collected for RNA extraction. RNA quality and concentration were determined spectrophotometrically by using the NanoDrop 2000 spectrophotometer (Thermo Scientific, Waltham, MA, USA). RNA integrity was evaluated using the Agilent 2100 Bioanalyzer (Agilent Technologies, Santa Clara, CA, USA) and examined after gel electrophoresis. Qualified samples were subjected to subsequent analysis. Libraries were constructed using TruSeq Stranded Total RNA with Ribo-Zero Gold (Illumina, San Diego, CA, USA) according to the manufacturer’s instructions. Then, these libraries were sequenced on the Illumina sequencing platform HiSeqTM 2500 (Illumina, USA). Two biological replicates were performed.

### 2.10. Transcriptome Sequencing, Analysis and qRT-PCR

Fastp software (version v0.23.3) was used to filter raw reads by quality, and STAR (version 2.7.10b) was used to align clean reads to the reference genome of *E. coli* BL21 (DE3). The expression abundance of each gene in each sample was identified by sequence similarity comparison. Stringtie was used to obtain the number of reads aligned to the gene in each sample and calculate gene count. The gene counts of each sample were normalized by DESeq2 software (version v2), and the *p*-value and fold change (FC) of difference comparison were calculated. The differentially expressed genes (DGEs) were screened based on false discovery rate (FDR) < 0.05 and FC > 2. Gene Ontology (GO)- and Kyoto Encyclopedia of Genes and Genomes (KEGG)-enriched analyses of the different genes were performed by clusterProfiler.

QRT-PCR analysis was performed by standard techniques. The primer sequences used are shown in Appendix A. *gyrA* and 16s rRNA genes were used as housekeeping genes.

### 2.11. Glycogen and Glucose Concentration Measurement

The intracellular glycogen and glucose concentration was measured using a Glycogen Content Assay Kit (Solarbio, Beijing, China) and Glucose Content Assay Kit (Solarbio, China). All operations were performed according to the instructions. Absorbance was measured using FlexStation3 (Molecular Devices, USA). Three biological replicates were performed for each treatment condition.

### 2.12. Statistical Analysis

All analyses were performed in triplicate. The experimental results were expressed as mean ± standard deviation. One-way analysis of variance (ANOVA) was performed to compare the significance level at (*p* < 0.05) using SPSS 13.0. All graphs and calculations were performed using Origin 2023b (Origin Lab Corporation, Northampton, MA, USA).

## 3. Results

### 3.1. HI-PMF Affects the Ratio of NADH and NAD^+^ in E. coli

As mentioned earlier, some studies reported damaging effects of HI-PMF on cells [38,39], while others insisted on there being no significant effects [40,41,52]. Thus, the growth rate and morphological variation in *E. coli* after being treated by HI-PMF at 2.5 T with 15 pulses were examined. The samples were resuspended in PBS and treated with HI-PMF. Then the tests were conducted immediately (Figure 1A). The results showed that the growth rate of *E. coli* did not vary significantly when compared to the control group (Appendix A). The cell morphology also showed no significant changes measured by SEM (Appendix A). Additionally, the electrical conductivity of the bacterial suspension remained the same before and after the treatment (Appendix A), indicating that no cells were broken. The above results all demonstrate that the treatment by HI-PMF at 2.5 T with 15 pulses may not have had a significant impact on the cell morphology.

Therefore, we further tested the cellular NADH/NAD^+^ ratio, a representative of cellular redox state that often changes in *E. coli* when subjected to external stimuli, via a method based on the SoNar fluorescent protein [53]. It is reported that the emission intensity of SoNar varies with NADH or NAD^+^ concentration in host cells; when these are excited at 405 nm or 485 nm, respectively, the intensity ratio of these two emissions represents the NADH/NAD^+^ ratio within the host cell [53]. After being treated by HI-PMF at 2.5 T for 15 pulses, the NADH/NAD^+^ ratio in *E. coli* cells was observed to be significantly increased (Figure 1B,C), suggesting an enhancement in carbon metabolism. Then, *E. coli* cells were also treated under magnetic fields ranging from 0 to 3.5 T with 10 pulses to explore the relationship between the NADH/NAD^+^ ratio and HI-PMF intensity. It was found that the NADH/NAD^+^ ratio increased with the increment of intensity of HI-PMF in the treatment (Figure 1D), meaning that a higher intensity of HI-PMF probably promoted a more effective carbon metabolism. Consequently, the number of pulses was then varied from 0 to 30, keeping the intensity constant at 2.5 T. As the number of pulses increased, the NADH/NAD^+^ ratio of the *E. coli* cells initially increased and peaked at 15 pulses, then decreased (Figure 1E), which might be due to the depletion of carbon sources in the cells. That is to say, HI-PMF treatment at 2.5 T could significantly change the carbon metabolism of *E. coli* cells.

### 3.2. HI-PMF Increases the Survival Rate of E. coli Cells by Promoting Their Division

To further assess the physiological effects of HI-PMF, the plate counting method was employed to calculate the cell number (Appendix A) and the survival rate (Figure 1F) of *E. coli* cells post-treatment. The survival rate exhibited a slight increase after treatment at 2.5 T with different pulses (Figure 1F). The most significant increase was observed at 5 pulses, showing an about 60% higher survival rate compared to the control. As the number of pulses increased, the increase in survival rate diminished and eventually stabilized at about 20%. The enhanced viability implies that 2.5 T HI-PMF treatment stimulates *E. coli* division.

This phenomenon was further confirmed by measuring cell length. After being treated by HI-PMF at 2.5 T and 15 pulses, the obtained cells were shorter and denser than those in the control group (Figure 1G,H), indicating that cell division was promoted. This result differs from those of the existing studies on HI-PMF [38,40], but is consistent with the studies demonstrating the growth-promoting effects of WI-MF [54,55]. The reasons for these differences are worth exploring.

### 3.3. Differentially Expressed Genes in E. coli After Being Treated by HI-PMF and Their Expression Profiles

To delve into the mechanisms behind the alterations in the NADH/NAD^+^ ratio and cell division rates, a transcriptome analysis of *E. coli* was conducted with and without HI-PMF treatment. Based on thresholds for FDR < 0.05 and log_2_ |fold change (FC)| ≥ 1, the transcriptome analysis revealed 1128 differentially expressed genes (DEGs) post-treatment (Figure 2A,D), with 920 DEGs downregulated and 208 DEGs upregulated.

KEGG analysis of upregulated genes showed the enriched genes were mainly related to oxidative phosphorylation, carbon metabolism, glycolysis, the citrate cycle, and the phosphotransferase system (Figure 2B). In particular, many genes involved in glycolysis were upregulated (Figure 2E,G), which was consistent with the observed increase in the NADH/NAD^+^ ratio. The genes responsible for glycogen degradation were also upregulated (Appendix A). These upregulated genes mean that *E. coli* increases carbon metabolism and energy production after being treated by HI-PMF. KEGG analysis of downregulated genes also showed that the enriched genes were mainly related to carbon metabolism processes, such as fructose and mannose metabolism, which was attributed to the regulation network of cellular carbon metabolism (Figure 2C). Among them, four glycolysis-related genes (*pgk*, *eno*, *acnB* and *mdh*) and three glycogen metabolism-related genes (*pgm*, *pgi* and *galU*) were selected and qRT-PCR was conducted to verify their expression. These genes displayed similar upregulation in qRT-PCR (Figure 2H) and transcriptome analysis (Figure 2I). Thus, transcriptome analysis proves that *E. coli* shows increases in carbon metabolism and energy production after being treated by HI-PMF.

Transcriptome analysis also revealed many cell-division-related genes were upregulated, which was probably an underlying reason for the observed increase in cell count, although some of them were below the standard of log2 |FC| ≥ 1 (Appendix A), causing the DEGs related to sieved cell division to tend to be a little low (Figure 2F). In addition, there may be other possible factors caused by HI-PMF that indirectly led to the increase in cell division. Similarly, three division-related genes (*ftsZ*, *ftsK* and *minD*) were selected for RT-qPCR testing and displayed similar upregulation in RT-qPCR (Figure 2H) and transcriptome analysis (Figure 2I).

MF is widely believed to induce cellular ROS through RPM [45,48]. But the DEGs related to ROS response were a little obscure. So, the expression levels of all redox-based transcriptional regulatory factors in *E. coli* were further examined [56]. Nearly all of them were upregulated after HI-PMF treatment (Appendix A). This indicates that *E. coli* also underwent a ROS response process during HI-PMF treatment. However, the ROS produced by 2.5 T HI-PMF treatment was moderate and was quickly eliminated by subsequent cell antioxidant processes, which was also confirmed by the increased NADH/NAD^+^ ratio observed later.

In addition, qRT-PCR validation of dozens of randomly selected genes from the transcriptome results all showed (Appendix A) the same trends as the transcriptome data (Appendix A).

Therefore, the transcriptome analyses further revealed the underlying gene regulation responsible for the increased NADH/NAD^+^ ratio and cell division.

### 3.4. HI-PMF Treatment Relieves Nutritional Inhibition on Cell Division of E. coli

From the above results, it can be seen that the mechanism for the increased cell division of *E. coli* during HI-PMF treatment was complex and needs further exploration. As reported, the cell division of *E. coli* is regulated by various mechanisms, the most one being nutritional inhibition [57]. This refers to the inhibition effect of a high intracellular glucose level on the cell division of *E. coli* [50]. After being treated by HI-PMF, increases in the carbon metabolism and its corresponding upregulated genes were tested. Therefore, it was speculated that the increased cell division of *E. coli* after being treated by HI-PMF was the result of enhanced glucose consumption and lowered intracellular glucose levels during the process. Glycogen is the glucose-storage reservoir in *E. coli* [58] and was monitored as a representation of the intracellular glucose level. It was found that the glycogen content in the treated group was much lower than that in the control group (Figure 3E), which supported our hypothesis.

For further validation, the *pgm* and *kilR* genes were selected to be knocked out. The *pgm* gene is crucial for the nutritional inhibition of cell division in *E. coli* [50], and the product of the *kilR* gene can also inhibit cell division [59]. The mutant strains and wild-type (wt) strains were both treated with 2.5 T HI-PMF and then tested. The deficient strain of *kilR* (Δ*kilR*) also showed an increase in the NADH/NAD^+^ ratio after HI-PMF treatment (Figure 3A,D), similar to the wt strain (Figure 3C,D), while Δ*pgm* led to a contrary effect (Figure 3B,D), caused by the lower glucose storage owing to its absence. On the other hand, knockout of both genes resulted in increased cell division, and the knockout strains had smaller sizes than the original strains in the stationary phase (Figure 3G). However, after HI-PMF treatment, the Δ*pgm* strain size did not further shorten, because *pgm* gene deficiency eliminated the nutritional inhibition of cell division, and HI-PMF could not promote cell division any further. However, the Δ*kilR* strain’s size further decreased after HI-PMF, indicating an overlapping effect occurred and that the *kilR* gene and HI-PMF treatment might promote different mechanisms for cell division in *E. coli*. Interestingly, the knockout of both genes could result in the disappearance of the increased survival rate formerly observed after 2.5 T HI-PMF treatment (Figure 3F), and *kilR* deletion even caused a reduction in survival rate. The *kilR* deletion might cause cell death under HI-PMF treatment as it as been reported that the *kilR* gene plays a key role in *E. coli* survival under stress [59].

Three DEGs (*glcE*, *eutK* and *prpR*), which relate to carbon metabolism and secondary carbon source utilization (Appendix A), were overexpressed to disturb carbon metabolism. All overexpressions led to a decrease in cell count in the stationary phase (Appendix A). The cellular glycogen levels in these three overexpressed strains were all lower than those in the original strains. The glycogen levels of the *eutK* and *prpR* overexpressed strains were further decreased after being treated by HI-PMF at 2.5 T and 15 pulses, while the glycogen level of the *glcE* overexpressed strain (Figure 3H) was initially the lowest and did not demonstrate further significant change after HI-PMF treatment. After metabolism disturbance, the nutritional inhibition of cell division was relieved and thus the mechanism for HI-PMF treatment on cell division disappeared, thus the survival rates of the *glcE*, *eutK,* and *prpR* overexpressed strains did not increase any further after HI-PMF treatment (Figure 3I).

These results revealed that the mechanism of nutritional inhibition was essential for the cell division promotion induced by HI-PMF treatment, which disappeared with *pgm* deletion or glucose metabolism disruption. However, the *kilR* gene is not directly related to the promotion of cell division but rather plays a key role in *E. coli* survival after being treated by HI-PMF.

### 3.5. Starvation Incubation Decreases the Survival of E. coli Under HI-PMF Treatment

In order to investigate the mechanism of nutritional inhibition, *E. coli* was suspended in PBS that did not contain glucose or other carbon sources during HI-PMF treatment. Therefore, these *E. coli* cells consumed their own glucose storage in this process, leading to the disappearance of division inhibition when glucose decreased to the threshold level. What would happen if glucose was supplied or glucose storage was consumed in advance? Further experiments were performed. In these experiments, part of the *E. coli* culture was resuspended in PBS for 2 h before HI-PMF treatment and was divided into two groups (Figure 4A). The starvation group (S group) did not receive any added glucose, while the other (S + G group) was supplied with 0.04% glucose. The residual culture was incubated in LB broth until treatment. Then, it was also divided into two groups and resuspended in PBS. One of them was set as the control group (C group) and received no additional glucose, and the other (C + G group) received 0.04% added glucose. All the groups were treated by 2.5T HI-PMF at 15 pulses and tested simultaneously (Figure 4A).

Without HI-PMF treatment, the NADH/NAD^+^ ratios (Figure 4F and Appendix A) and cellular glycogen concentrations (Figure 4H) ranked in the following order: S + G groups > C + G group > C group > H group. The cell sizes (Figure 4G) and cell counts (Figure 4I) exhibited more complex results. S group had the highest cell count and the shortest cell length, suggesting a loss of nutritional inhibition. The cell sizes and cell counts of the S + G group and C group were roughly similar, but both were lower than those of C + G group. We speculate that the sudden rise in glucose levels may have stimulated cell growth and division in the C + G group, whereas some cells may have died during the incubation process in the S + G group.

After HI-PMF treatment at 2.5 T and 15 pulses, the NADH/NAD^+^ ratio of the S group decreased (Figure 4F), while that of the C group increased as before, and the remaining two groups did not change significantly. Consistent with the changes in the NADH/NAD^+^ ratio, the cellular glycogen concentration of the S group was too low to decrease further after HI-PMF, while the C group consumed part of its glycogen (Figure 4H), and the remaining two groups did not vary significantly. Intracellular glucose concentrations of these four groups were also measured to test the absorption of glucose by the *E. coli* cells (Appendix A). The two groups supplied with glucose had more intracellular glucose content than the remaining two groups, and the C + G group absorbed more glucose during HI-PMF treatment, which was consistent with the regulation of the carbohydrate transport system revealed by transcriptome analyses.

The influence on cell size and cell count by nutritional status during HI-PMF treatment was observed. For the S group, the newly divided cells were rather fragile, leading to a reduction in cell count after being treated by HI-PMF (Figure 4I), but the cell lengths did not change (Figure 4G). In contrast, the S + G group showed no change in cell length (Figure 4G) or cell count (Figure 4I) due to the inhibition of division by glucose and stronger stress resistance. The increase in cell count (Figure 4I) and decrease in cell length (Figure 4G) in the C group indicated the relief of division inhibition. Both the cell length (Figure 4G) and cell count (Figure 4I) of the C + G group were slightly decreased, probably due to the death of newly divided cells caused by the treatment of HI-PMF.

In general, *E. coli* cells increased their glucose metabolism to resist the stress from the HI-PMF treatment, and their survival rate under HI-PMF treatment depended on their nutritional status in situ, i.e., the initial level of intracellular glucose before the treatment or extracellular glucose availability during the treatment.

### 3.6. ROS Participates in the Response of E. coli Cells to HI-PMF Treatment

As previously shown by the transcriptome analysis, the genes related to ROS elimination were upregulated (Appendix A). Meanwhile, as reported in the literature, ROS stimulation can also regulate cellular metabolism [60]. Therefore, it is supposed that the increased glucose consumption may be related to ROS production caused by the HI-PMF treatment. To further validate this supposition, the *soxR* gene, which controls the transcription of the regulon involved in cell defense against ROS, was knocked out or overexpressed. The result showed that the Δ*soxR* strain was more sensitive to HI-PMF treatment (Figure 5A), while the overexpressed strain did not respond significantly. On the other hand, Tert-butylalcohol (TBA), a free radical scavenger, was added to the treated buffer in the wt group at a concentration of 10mM. It can assist cells to resist ROS [61] and does not affect cell metabolism [62,63]. This group also did not exhibit a significant response to HI-PMF treatment (Figure 5A). Furthermore, the tested H_2_O_2_ concentration of the Δ*soxR* strain was increased with HI-PMF treatment due to its deficiency in ROS resistance, while all other groups did not demonstrate any change in the H_2_O_2_ concentration (Figure 5B).

In addition, the NADH/NAD^+^ ratio and glycogen concentration of these groups also verified the supposition. The NADH/NAD^+^ ratio of the Δ*soxR* strain decreased significantly (Figure 5E) and that of the wt strain increased markedly after HI-PMF treatment (Figure 5C). However, the glycogen concentrations of the Δ*soxR* strain and wt strain both decreased significantly during this process (Figure 5H). And the remaining two groups remained unchanged in either aspect (Figure 5D,F,H), owing to their pre-existing ability to resist ROS before HI-PMF treatment. These results indicate that ROS participates in the response of *E. coli* cells to HI-PMF treatment, and the *soxR* gene is directly involved in this process.

### 3.7. Appropriate ROS Stimulation Can Promote E. coli Cell Division

So far, the relationship between ROS stimulation and the intracellular NADH/NAD^+^ ratio and cell division is not well understood. To address this issue, *E. coli* was treated with various concentrations of H_2_O_2_. Using a microplate reader, the NADH/NAD^+^ ratios of all groups were continuously monitored by the fluorescence ratio of SoNar under H_2_O_2_ treatment (Figure 6A). It should be pointed out that all groups exhibited similar NADH/NAD^+^ ratios before H_2_O_2_ treatment. After the addition of H_2_O_2_, the NADH/NAD^+^ ratio in all groups immediately declined, regardless of any concentrations of H_2_O_2_ (Figure 6A). In the control group, sterile water was added instead, and the NADH/NAD^+^ ratio stabilized around 0.96 after an initial slight reduction. In contrast, the treated groups displayed several distinct peaks in the NADH/NAD^+^ ratio over time after the initial drop (Figure 6A). For instance, in the group treated with 25 mM H_2_O_2_, the ratio first declined to 0.85, then recovered to a peak of 0.97 at 27 min, then dropped again to 0.88 at 30 min, gradually rebounded to another peak of 1.01 at 60 min, and then declined to 0.94, after which it basically stabilized (Figure 6A). As the concentration of H_2_O_2_ increased, the amplitudes of these peaks decreased, and the times that their first peaks occurred were delayed. However, the next peak consistently appeared at approximately 55 min (Figure 6A). These results indicate that exposure to an appropriate concentration of H_2_O_2_ can induce a temporary increase in the NADH/NAD^+^ ratio in *E. coli* cells within a defined timeframe. The samples treated for 60 min were also observed by microscope (Figure 6C), and the results were consistent with those measured by microplate reader.

Furthermore, cell counts were also analyzed following H_2_O_2_ treatment. The group treated with 25 mM H_2_O_2_ showed an initial decline at 20 min, followed by an increase at 40 min, a pattern roughly consistent with the response observed after HI-PMF treatment, while the control group exhibited a gradual decrease in cell number over time (Figure 6B). In contrast, the cell counts in groups treated with 75 mM and 100 mM continuously declined (Figure 6B). Interestingly, the group treated with 50 mM showed an immediate increase in cell count at 20 min followed by a decline at 40 min (Figure 6B). The cell lengths measured at 60 min showed that those of the groups treated with 25mM and 50 mM were shortened, which may be due to cell division (Figure 6E). But the group treated with 100mM was also shortened, which might be due to cell shrinking under a high concentration of H_2_O_2_ (Figure 6E).

Further investigation of the effect of moderate H_2_O_2_ concentrations (40–65 mM) obtained two distinct response patterns. Cell count dynamics of groups treated with 40 mM and 45 mM H_2_O_2_ were similar to those observed with 25 mM treatment (Appendix A). In contrast, those groups treated with 55 mM, 60 mM, and 65 mM followed a similar pattern to that of the 50 mM group (Appendix A). Notably, in all groups, cell counts generally decreased as H_2_O_2_ concentration increased. This phenomenon aligns with previous reports indicating that moderate H_2_O_2_ concentrations are less lethal to *E. coli* cells than both lower and higher concentrations [64], which is also reflected in the survival rates measured at 20 min after treatment across the H_2_O_2_ gradient (Appendix A). It is possible that the combination of low lethality and a transient promotion of cell division in the 50 mM to 65 mM range of H_2_O_2_ contributed in some way to the increased cell counts observed at 20 min.

The results suggest that appropriate ROS stimulation can enhance the cell metabolism of *E. coli*, transiently elevate NADH levels, relieve glucose-mediated inhibition of cell division, and subsequently increase the cell number, a response that is strikingly similar to that observed under the 2.5 T HI-PMF treatment of *E. coli* cells. These findings imply that the cellular ROS induced by 2.5 T HI-PMF treatment may also fall within a moderate range for *E. coli* cells, promoting cell division.

## 4. Discussion

As mentioned earlier, some studies have disclosed the damaging effects of HI-PMF on cells [38,39], while others have reported no/little harm [40,41,52]. However, this study observed both phenomena, as well as a third one: the promotion of *E. coli* cell division. The effects resulted from the initial cellular glucose level of *E. coli.* The HI-PMF we applied induced ROS stimuli that further promoted cellular metabolism. In glucose-free solutions, cells consumed their own glucose storage, which relieved the division inhibition imposed by high-level glucose during HI-PMF treatment (Figure 7B). On the contrary, cells with a glucose level that was too low did not have enough energy to cope with HI-PMF stress, resulting in partial cell death (Figure 7D), while in solutions containing glucose, cells had a sufficient glucose supply to maintain the division inhibition during HI-PMF treatment (Figure 7C). Throughout these processes, the NADH/NAD^+^ ratio also fluctuated corresponding to the variations in cellular glucose levels. Therefore, cellular glucose level, often overlooked in previous studies, plays a key role in these different outcomes. Similar phenomena have been reported in the electric stimulation of cells, where an identical electrical stimulus can produce opposite polarization dynamics depending on the cell’s state [65].

This study used stationary-phase *E. coli* cells and found a new phenomenon, that HI-PMF promoted *E. coli* cells division. For stationary-phase *E. coli* cells, a phenomenon called “reductive division” has been previously reported [66,67]. Reductive division refers to cell division in the absence of further growth upon entry of cells into stationary phase, which was thought to result from chromosome replication during the stationary phase [66]. In another study, reductive division of *E. coli* was also reported after 28–52 h of cultivation with glucose nearly fully consumed, which led to a reduction in cell size [67]. These phenomena are similar to the division observed after HI-PMF and H_2_O_2_ treatments in this study, suggesting that the relief of division inhibition by lowering cellular glucose level may be one of the factors contributing to reductive division.

In another study, the transcriptome of *L. monocytogenes* was analyzed after being treated with HI-PMF at 8 T and 20 pulses [39]. The gene ontology (GO) analysis of this transcriptome reveals that the five biological processes with the largest number of DEGs are the metabolic process, cellular process, establishment of localization, location, and response to stimulus. The KEGG pathway classifications of DEGs with the top five rates are carbohydrate metabolism, membrane transport, energy metabolism, global and overview maps, and amino acid metabolism. These findings demonstrate that HI-PMF influences cellular carbon metabolism, despite differences in experimental setups compared to our study.

It is well-established that MFs promote intracellular ROS production, leading to various physiological changes [47,68,69]. It is therefore reasonable to assume that ROS production by HI-PMF is responsible for the observed effects. This study confirms the involvement of ROS in the biological effects of HI-PMF on *E. coli*, and highlights that the outcomes of HI-PMF treatment can vary depending on the cellular glucose levels, which are influenced by the experimental conditions. In other words, the different effects of HI-PMF on cells may be due to the varying anti-ROS capacities of different cells, which depend on the cell types, physiological states and experimental parameters. Moreover, direct ROS stimulation also has this effect. It is worth considering whether other stimuli have similar effects.

This study demonstrates that HI-PMF can enhance glucose metabolism in *E. coli* and increase cell number in the stationary phase under specific conditions. Leveraging this mechanism in *E. coli* fermentation may improve both biomass and substrate utilization during this phase. Furthermore, HI-PMF affects the intercellular NADH/NAD^+^ ratio, which may facilitate the synthesis of valuable metabolites. Nevertheless, the optimal treatment duration and MF intensity require further investigation. Although previous studies have established that WI-MFs can enhance biomass and product yields in microbial fermentation, they typically apply magnetic exposure from the early fermentation stage [13,70]. While this approach accelerates growth, it often leads to premature decline and requires prolonged treatment [71]. In contrast, the present method utilizes short-term HI-PMF exposure specifically during the stationary phase, thereby avoiding these limitations.

Some studies have also explored the effects of MFs on enzymatic activity [72], cell membrane [73], and membrane potential [26,44]. Compared with weak dynamic MFs, HI-PMF treatment has a higher intensity and faster change rate, providing a stronger stimulus to cell membrane and membrane potential. Since no significant membrane damage was observed in this study, we did not delve into this aspect. However, recent studies have suggested that bacterial membrane potential plays an important role in bacterial growth and metabolism [74,75]. Some physiological phenomena have yet to be fully explored. Further research might be conducted in these directions in the future.

## 5. Conclusions

The phenotype and physiological variations in *E. coli* were examined after being treated by 2.5 T HI-PMF, observing notable increases in NADH/NAD^+^ level and survival rate. Transcriptome analysis revealed significant effects on cellular glucose metabolism. It was also found that glucose consumption in this process led to the alleviation of cell division inhibition, resulting in the increase in survival rate. That is to say, the initial intracellular glycose level might finally determine the survival rate of *E. coli* under HI-PMF treatment. Furthermore, ROS production caused by HI-PMF was proved to be the main reason attributed to these physiological variations, as direct ROS treatment induced similar effects as could be seen after HI-PMF treatment. Therefore, this study deepens the understanding of the biological effects of HI-PMF. The underlying mechanism is likely that HI-PMF treatment promotes ROS production, then enhances cellular glucose metabolism, which further influences cell division and survival rates according to the initial intracellular glucose level.

## Figures and Tables

**Figure 1 biomolecules-15-01550-f001:**
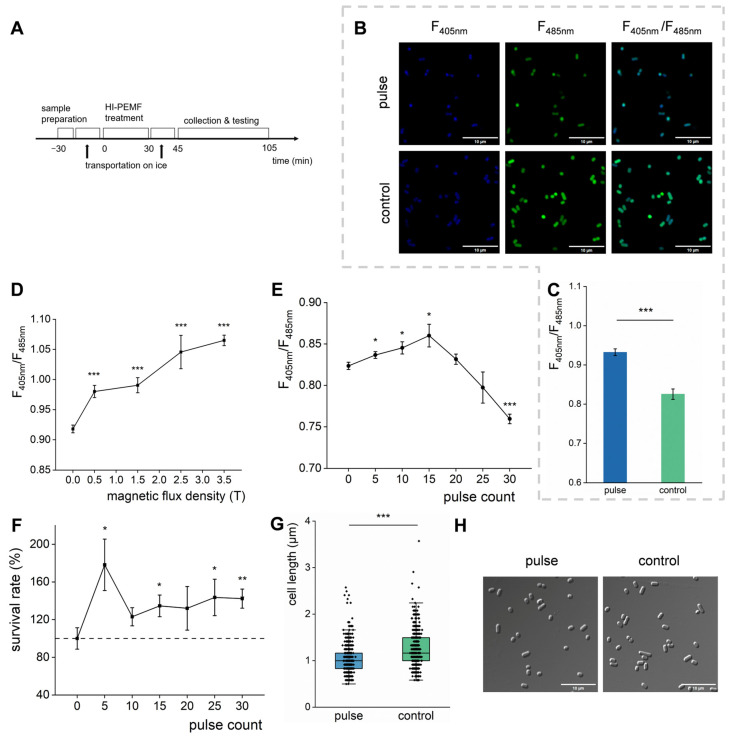
The changes in *E. coli* phenotype after being treated by HI-PMF. (**A**) The schedule of HI-PMF treatment and testing. (**B**) The fluorescence image of *E. coli* expressing SoNar with/without treatment. F_405nm_, captured under 405 nm excitation and 520 nm emission. F_485nm_, captured under 485 nm excitation and 520 nm emission. F_405nm_/F_485nm_, merge of F_405nm_ and F_485nm_. (**C**) The NADH/NAD^+^ ratio of *E. coli* tested by the fluorescence of SoNar. (**D**) The changes in NADH/NAD^+^ by intensity of HI-PMF. (**E**) The changes in NADH/NAD^+^ by pulse count of HI-PMF at 2.5 T. (**F**) The changes in the survival rate by pulse count of 2.5 T HI-PMF. The dashed line indicates the survival rate as a percentage. (**G**) The cell length with/without 2.5 T and 15 pulses HI-PMF treatment. (**H**) Micrograph of *E. coli* with (left) or without (right) HI-PMF treatment. * *p* < 0.05, ** *p* < 0.01, *** *p* < 0.001. Data are all expressed as mean ± SD (*n* = 3 except (**G**) and *n* = 500 in (**G**)).

**Figure 2 biomolecules-15-01550-f002:**
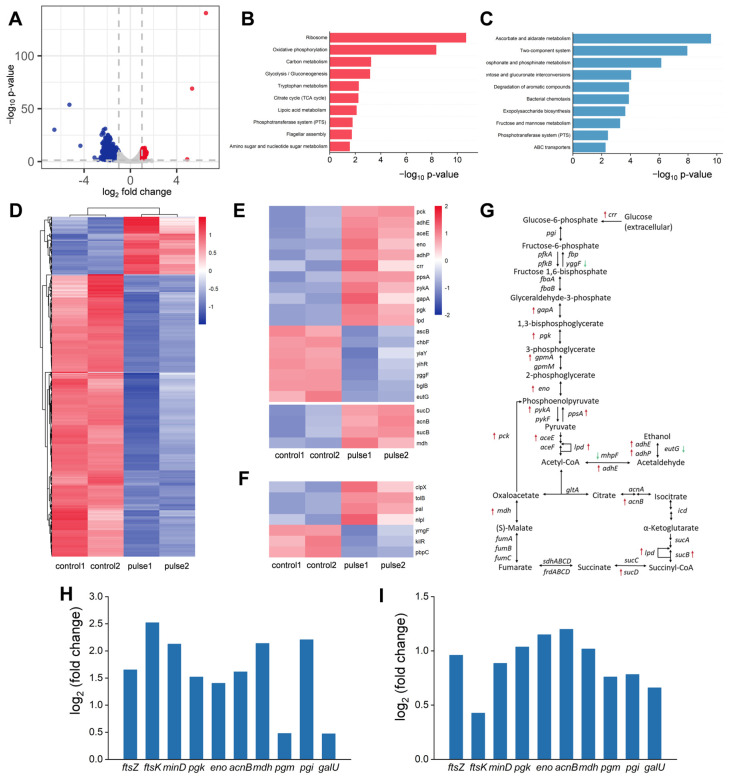
The DEGs of *E. coli* with or without HI-PMF treatment. (**A**) The volcano plot of the DEGs. The dashed line shows the thresholds for FDR < 0.05 and log fold change |FC| ≥ 1. (**B**) KEGG-enriched analysis of upregulated genes. (**C**) KEGG-enriched analysis of downregulated genes. (**D**) The heat map of relative expression levels of the DEGs. (**E**) Difference in expression of genes related to glycolysis and the TCA cycle. (**F**) Difference in expression of genes related to cell division. (**G**) A diagrammatic sketch of glycolysis and the TCA cycle. The upregulated genes are marked by red upward arrows, while the downregulated genes are marked by green down arrows. (**H**) The changes in gene expression related to cell division and glucose metabolism measured by qRT-PCR. (**I**) The changes in gene expression related to cell division and glucose metabolism measured by transcriptome sequencing (*n* = 2 expect (**H**), *n* = 4 in (**H**)).

**Figure 3 biomolecules-15-01550-f003:**
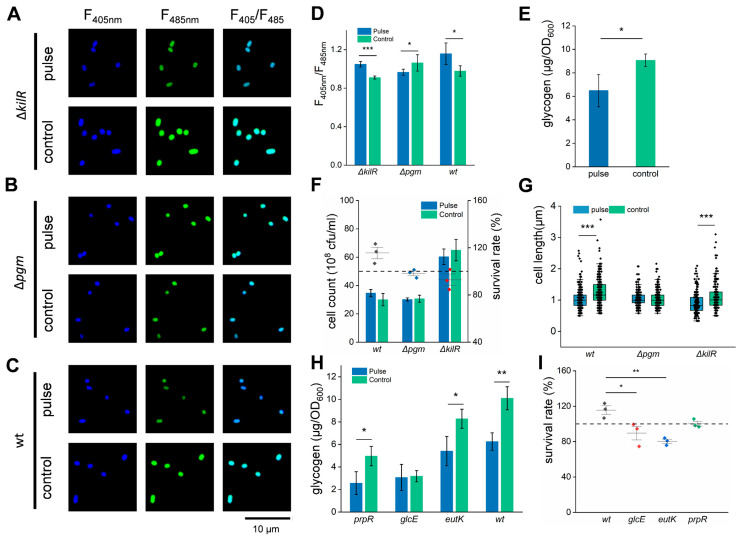
The release of nutritional inhibition after being treated by HI-PMF. A-C The fluorescence image of *E. coli* expressing SoNar with/without treatment, which was the knock out of *kilR* (**A**), knock out of *pgm* (**B**) or original strain (**C**). (**D**) The NADH/NAD^+^ ratio of different strains tested by the fluorescence of SoNar, quantified from fluorescent images. (**E**) The changes in cellular glycogen concentration with (pulse) or without (control) HI-PMF treatment. (**F**) The cell count (column graph) and the survival rates (interval plot) of gene knockout strains after being treated by HI-PMF. Dash line represents a survival rate of 100%. (**G**) The cell lengths of gene knockout strains with/without HI-PMF treatment. Δ*pgm*, *n* = 300, Δ*kilR*, *n* = 200, wt, *n* = 300. (**H**) The glycogen concentration of gene overexpression strains with/without HI-PMF treatment. (**I**) The survival rates of *E. coli* with gene overexpression after being treated by HI-PMF. The dashed line represents a survival rate of 100%. wt, wild type group of *E. coli* BL21 (DE3) expressing SoNar. * *p* < 0.05. ** *p* < 0.01, *** *p* < 0.001. Data were all expressed as mean ± SD (*n* = 3 except (**G**)).

**Figure 4 biomolecules-15-01550-f004:**
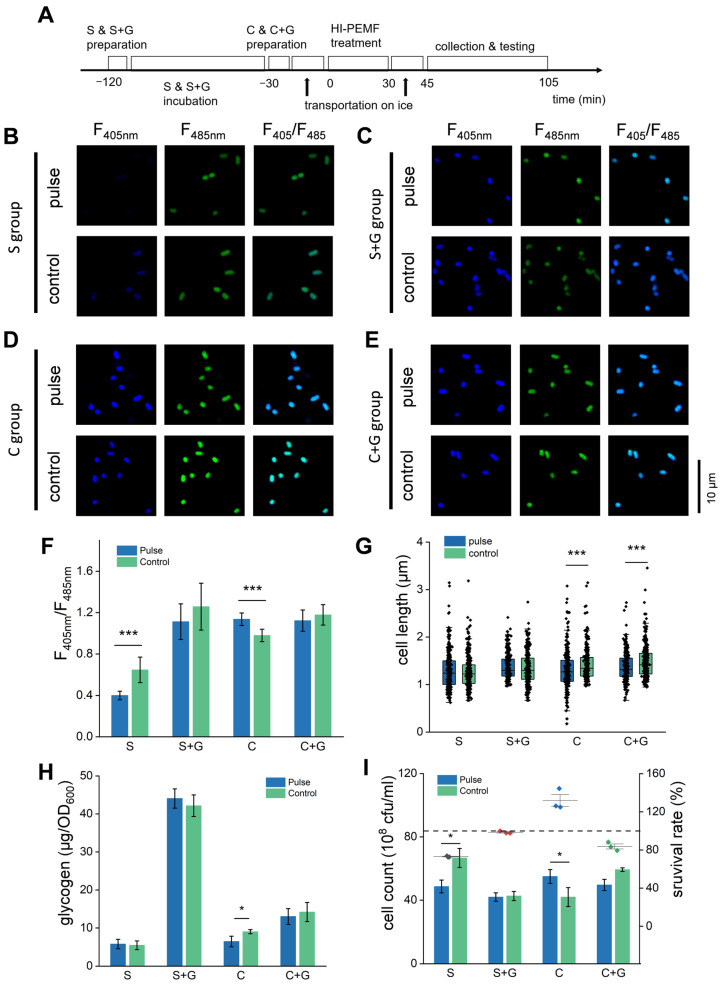
The influence of initial cellular glucose on changes in *E. coli* after being treated by HI-PMF. (**A**) The schedule of sample preparation, HI-PMF treatment, and testing. (**B**–**E**) The fluorescence image of *E. coli* expressing SoNar with/without treatment, which was S group (**B**), S + G group (**C**), C group (**D**) and C + G group (**E**). (**F**) The NADH/NAD^+^ ratio of different groups with/without HI-PMF treatment, quantified from fluorescent images. (**G**) The cell length of different groups with/without HI-PMF treatment (*n* = 300) (**H**) The cellular glycogen concentration with/without HI-PMF treatment. (**I**) The cell count (column graph) and the survival rates (interval plot) of different groups after being treated by HI-PMF. The dashed line represents a survival rate of 100%. S group: group retreated with PBS and no additional glucose for 2 h. S + G group: group retreated with PBS and 0.04% glucose for 2 h. C group: group without retreatment and additional glucose. C + G group: group without retreatment but receiving 0.04% glucose when treated by HI-PMF. * *p* < 0.05, *** *p* < 0.001. Data were all expressed as mean ± SD (*n* = 3 expect (**G**)).

**Figure 5 biomolecules-15-01550-f005:**
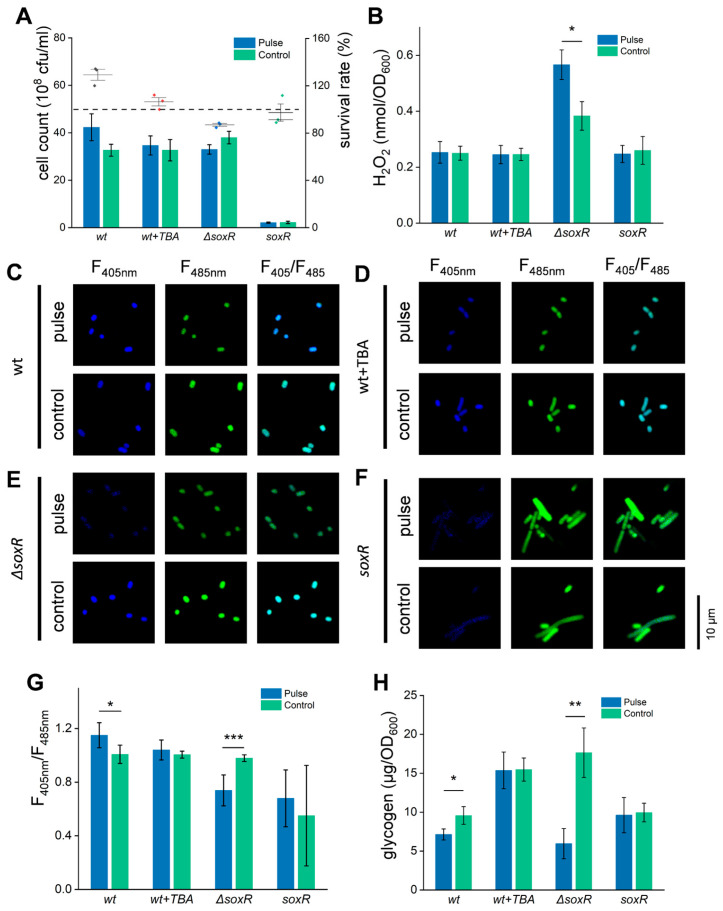
The participation of ROS in the cellular effects of HI-PMF (**A**) The cell count (column graph) and the survival rates (interval plot) of the wt group, the wt treated with TBA (tert-butylalcohol) group, the Δ*soxR* group, and the *soxR* overexpressing group with/without HI-PMF treatment. (**B**) The intracellular H_2_O_2_ concentration of the different groups with/without HI-PMF treatment. (**C**–**F**) The fluorescence image of *E. coli* expressing SoNar with/without treatment, which was the wt group (**C**), the wt treated with TBA group (**D**), the Δ*soxR* group (**E**) and the *soxR* overexpressing group (**F**). (**G**) The changes in NADH/NAD^+^ of different groups, quantified from fluorescent images. (**H**) The intracellular glycogen concentration of the different groups with/without HI-PMF treatment. * *p* < 0.05, ** *p* < 0.01, *** *p* < 0.001. Data were all expressed as mean ± SD (*n* = 3).

**Figure 6 biomolecules-15-01550-f006:**
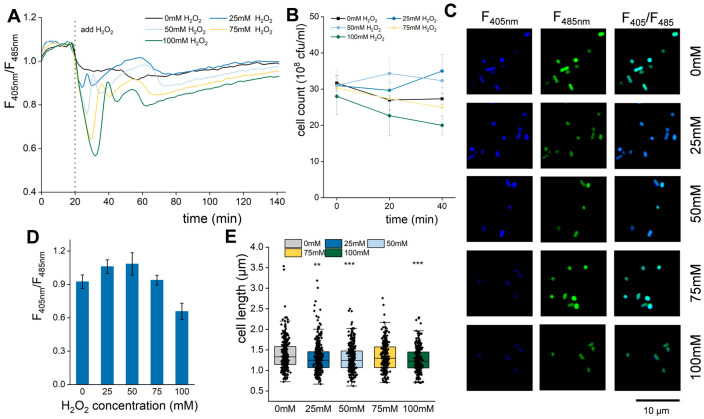
The influence of ROS stimulation on *E. coli* metabolism and division (**A**) The changes in NADH/NAD^+^ after treatment with different concentrations of H_2_O_2_. (**B**) The changes in cell count after treatment with different concentrations of H_2_O_2_. (**C**) The fluorescence image of *E. coli* treated with different concentrations of H_2_O_2_ for 60 min. (**D**) The NADH/NAD^+^ of *E. coli* treated with different concentrations of H_2_O_2_ for 60 min, quantified from fluorescent images. (**E**) the cell length of *E. coli* treated with different concentrations of H_2_O_2_ for 60 min. ** *p* < 0.01, *** *p* < 0.001. Data were all expressed as mean ± SD (*n* = 3 expect (**D**,**E**)).

**Figure 7 biomolecules-15-01550-f007:**
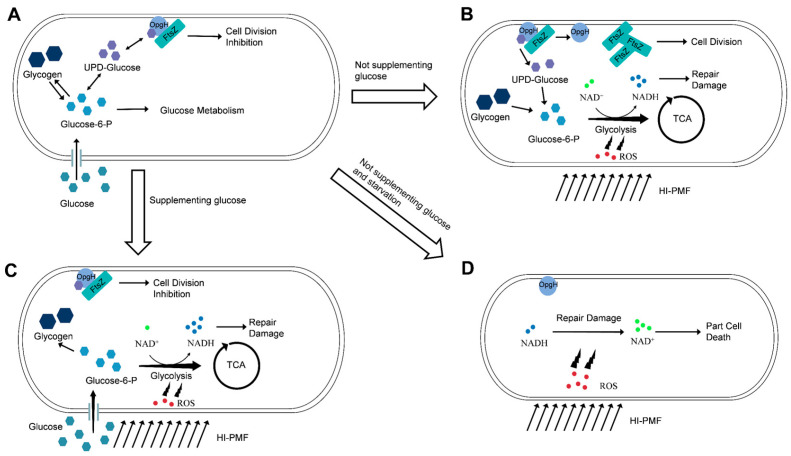
The diagrammatic sketch of *E. coli* in response to HI-PMF under different conditions. (**A**). *E. coli* under normal circumstances. (**B**). *E. coli* in response to HI-PMF without supplementing glucose. (**C**). *E. coli* response to HI-PMF when supplementing glucose. (**D**). *E. coli* in response to HI-PMF without supplementing glucose and inducing starvation.

## Data Availability

The transcriptome data generated in this study for Figure 2 have been deposited at Gene Expression Omnibus under accession codes GSE278595. Raw data generated during the current study have been deposited at Zenodo under DOI 10.5281/zenodo.17293788.

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
