# Peer review of "High-Intensity Pulse Magnetic Fields Affect Redox Homeostasis and Survival Rate of Escherichia coli According to Initial Level of Intracellular Glucose"

_biomolecules, 2025, doi:10.3390/biom15111550_

Round 1
Reviewer 1 Report
Comments and Suggestions for Authors
Excellent article. But I'll start right away with the formatting issues that stand out: line 339, 576 – a space is missing before a square bracket; the hydrogen peroxide experiment (lines 125-132) – it's better to reduce the number of 2s in the hydrogen peroxide formula like in line 548. In figures 2 and 6, the first images are very small. Please enlarge them – it's impossible to see the meanings. I understand there are a lot of figures and I'd like to combine them, but I want this combination to not degrade the quality of perception. A very interesting study was conducted on the biological effects of a magnetic device on Escherichia coli and the corresponding physiological changes. The experiment on unlocking the intracellular NADH/NAD+ ratio, combined with improved cell survival, was excellent. The transcriptome analysis of glucose metabolism was a very important and well-designed experiment. The subsequent experiments were also very well designed. I was missing an explanation of how this technology might be applied in the future. If possible, please add that.
Author Response
Comment 1: line 339, 576 – a space is missing before a square bracket; the hydrogen peroxide experiment (lines 125-132) – it's better to reduce the number of 2s in the hydrogen peroxide formula like in line 548.
Response: Thank you! We have corrected the above mistakes in the revised manuscript.
Comment 2: In figures 2 and 6, the first images are very small. Please enlarge them – it's impossible to see the meanings. I understand there are a lot of figures and I'd like to combine them, but I want this combination to not degrade the quality of perception.
Response: Thank you! We have implemented these corrections. The changed pictures are as follows:
Figures 2:

Figures 6:

Comment 3: I was missing an explanation of how this technology might be applied in the future. If possible, please add that.
Response: Thanks for your valuable suggestion. We have expanded this point to highlight the application potential of this technology. A new paragraph has been incorporated into the Discussion section (lines 579-591 in the revised manuscript). The details are as follows:
lines 579-591: This study demonstrates that HI-PMF can enhance glucose metabolism in E. coli and increase cell number in the stationary phase under specific conditions. Leveraging this mechanism in E. coli fermentation may improve both biomass and substrate utilization during this phase. Furthermore, HI-PMF affects the intercellular NADH/NAD+ ratio, which may facilitate the synthesis of valuable metabolites. Nevertheless, the optimal treatment duration and MF intensity require further investigation. Although previous studies have established that WI-MFs can enhance biomass and product yields in microbial fermentation, they typically apply magnetic exposure from the early fermentation stage [13,70]. While this approach accelerates growth, it often leads to premature decline and requires prolonged treatment [71]. In contrast, the present method utilizes short-term HI-PMF exposure specifically during the stationary phase, thereby avoiding these limitations.
Reviewer 2 Report
Comments and Suggestions for Authors
Dear Authors,
Your manuscript entitled “High-intensity Pulse Magnetic Field Affects Escherichia coli Redox Homeostasis and Survival Rate According to the Initial Level of Intracellular Glucose” presents an interesting and rarely explored topic, namely the biological impact of high-intensity pulsed magnetic fields (HI-PMFs) on bacterial physiology. The manuscript is well organized and the experimental design appears solid.
The following suggestions are provided to further improve clarity, accuracy, and scientific consistency:
Lines 65-89: The manuscript would benefit from a clearer statement of the mechanistic hypothesis at the end of the Introduction—specifically, that HI-PMF is proposed to modulate redox balance and survival through glucose-dependent ROS regulation.
Lines 105-114: The justification for selecting 2.5 T and 15 pulses as the main HI-PMF condition is missing. Explain whether these parameters were chosen based on previous literature, preliminary tests, or equipment limitations.
Lines 168-169: The RNA sequencing analysis was performed with two biological replicates per group. While this number can still yield indicative results, it would be helpful to specify whether technical replicates or validation assays (e.g., qRT-PCR) were used to support these findings. Clarifying this point would enhance methodological transparency.
Sections 3.6-3.7: The results suggest that ROS generation correlates with glucose consumption and survival under HI-PMF. To strengthen this conclusion, briefly discuss whether scavenger experiments (TBA) affected NADH/NAD⁺ ratios or metabolic intermediates.
References: Check journal abbreviations and ensure consistent DOI formatting in accordance with Biomolecules style.
I think that the manuscript could be suitable for publication after minor revisions addressing the points above.
Author Response
Comment 1: Lines 65-89: The manuscript would benefit from a clearer statement of the mechanistic hypothesis at the end of the Introduction—specifically, that HI-PMF is proposed to modulate redox balance and survival through glucose-dependent ROS regulation.
Response: We are grateful to your valuable comment. The suggested content has been added to the Introduction. The related statements are as follows:
Lines 87-92: Finally, it was discovered that the increased metabolism was induced by the reactive oxygen species (ROS) generated by RPM, and appropriate ROS stimulation by H2O2 also could promote cell division of E. coli. Our study suggests that HI-PMF treatment enhances cellular glucose metabolism by promoting ROS production, and further affects cell survival according to the initial level of intracellular glucose.
Comment 2: Lines 105-114: The justification for selecting 2.5 T and 15 pulses as the main HI-PMF condition is missing. Explain whether these parameters were chosen based on previous literature, preliminary tests, or equipment limitations.
Response: Thanks for your comment. The parameters of HI-PMF were chosen based on preliminary tests and equipment limitations. Preliminary tests showed that 10 pulses HI-PMF treatment with intensities of 0.5 T, 1.5 T, 2.5 T and 3.5 T all led to the increase of E. coli NADH/NAD+. But the coli used in this study was unstable and overheated under HI-PMF at too high intensity and too many pulses. Therefore, 2.5 T was chosen as the main intensity. The number of pulses was then varied from 0 to 30 keeping the intensity constant at 2.5 T. As the number of pulses increased, the NADH/NAD+ ratio of E. coli cells initially increased and peaked at 15 pulses, then decreased. 15 pulses seemed to be a key point and was chosen as the main pulse number. The related data was showed at Fig. 1D, 1E, and we have now expanded the related statement in Section 2.2 as follows:
Lines 108-119: 2.2 HI-PMF treatment
Pulsed magnetic field equipment was made by school of electrical and electronic engineering, Huazhong University of Science and Technology. Fully charged capacitors discharge current to the treatment chamber made of coil to generate MF. The maximum MF intensity generated by the coli is 3.5 T. The stimulus waveform was monophasic, with a pulse width of 6 ms and 30 seconds interval (Fig. S1). It was paused for 5 minutes every 3 pulses to maintain the temperature at 20 degrees Celsius. The treatment chamber was cylindrical with 30 cm in height and 60 cm in diameter. Samples were placed inside the coil at 15 cm in height and 20 cm in diameter. The intensity of PMF of the treatment chamber is spatially uniform and adjusted by the input voltage. 10 pulses HI-PMF treatment with intensities of 0.5 T, 1.5 T, 2.5 T and 3.5 T were processed firstly. Then 2.5 T HI-PMF with pulse numbers from 5 to 30 were tested. Finally, 2.5 T and 15 pulses were chosen as main HI-PMF parameters and used in subsequent experiments.
Comment 3: Lines 168-169: The RNA sequencing analysis was performed with two biological replicates per group. While this number can still yield indicative results, it would be helpful to specify whether technical replicates or validation assays (e.g., qRT-PCR) were used to support these findings. Clarifying this point would enhance methodological transparency.
Response: Thanks for raising this important point. We agree that two biological replicates per group may produce errors, and have conducted qRT-PCR to support these findings. This aspect has been addressed in section 3.3. The related data are presented at Fig. 2H, 2I, S3C and S3D. The corresponding methods are attached in Section 2.10, and may be easy to be ignored due to it is not mentioned in the subheading. Relevant contents have been modified as follows:
Lines 175-188: 2.10 Transcriptome sequencing, analysis and qRT-PCR
Fastp software was used to filter raw reads by quality, and STAR was used to align clean reads to the reference genome of E. coli BL21 (DE3). The expression abundance of each gene in each sample was identified by sequence similarity comparison. Stringtie was used to obtain the number of reads aligned to the gene in each sample and calcu-late gene count. The number of gene count of each sample was normalized by DESeq2 software, when p-value and fold change (FC) of difference comparison were calcu-lated. The differentially expressed genes (DGEs) were screened based on false discov-ery rate (FDR) < 0.05 and FC > 2. The Gene Ontology (GO) and Kyoto Encyclopedia of Genes and Genomes (KEGG) enrichment analysis of the differential genes were per-formed by clusterProfiler.
QRT-PCR analysis was performed by standard techniques. The primer sequences used was in supplementary Table S2. The gyrA and 16s rRNA gene were used as house-keeping gene.
Comment 4: Sections 3.6-3.7: The results suggest that ROS generation correlates with glucose consumption and survival under HI-PMF. To strengthen this conclusion, briefly discuss whether scavenger experiments (TBA) affected NADH/NAD⁺ ratios or metabolic intermediates.
Response: Thanks for your valuable suggestion. The effects of tert-butylalcohol (TBA) on NADH/NAD⁺ ratios or metabolic intermediates should be discussed. TBA is a hard-metabolizable alcohol and accumulates as an intermediate metabolite in the bacteria capable of degrading methyl tert-butyl ether [62]. On the other hand, the TBA concentration used in this study is 10 mmol/L (0.00074 g/ml), a low and safe concentration for E. coli [63]. Therefore, TBA did not affect NADH/NAD⁺ ratios or metabolic intermediates in this study. To illustrate this, we revise the article as follows:
Lines 444-451: The result showed that ΔsoxR strain was more sensitive to HI-PMF treatment (Fig. 5A), while the overexpressed strain did not respond significantly. On the other hand, Tert-butylalcohol (TBA), a free radical scavenger, was added to treated buffer in wt group at a concentration of 10mM. It can assist cell to resist ROS [61] and does not affect cell metabolism [62,63]. This group also didn’t exhibit a significant response to HI-PMF treatment (Fig. 5A). Furthermore, the tested H2O2 concentration of the ΔsoxR strain was increased with HI-PMF treatment, due to its deficiency in ROS resistance, while all other groups did not measure any change in the H2O2 concentration (Fig. 5B).
Comment 5: References: Check journal abbreviations and ensure consistent DOI formatting in accordance with Biomolecules style.
Response: Thanks for your comments. We have checked all the references and modified their formats according to Biomolecules style.